# Adaptive Model for Biofeedback Data Flows Management in the Design of Interactive Immersive Environments

**Paulo Veloso Gomes** [1,*], **António Marques** [1], **João Donga** [2], **Catarina Sá** [1], **António Correia** [1] and **Javier Pereira** [3]

1. LabRP-CIR, Psychosocial Rehabilitation Laboratory, Center for Rehabilitation Research, School of Health, Polytechnic Institute of Porto, 4200-072 Porto, Portugal; ajmarques@ess.ipp.pt (A.M.); catarina.sa.12@gmail.com (C.S.); amsc@ess.ipp.pt (A.C.)
2. LabRP-CIR, Psychosocial Rehabilitation Laboratory, School of Media Arts and Design, Polytechnic Institute of Porto, 4480-876 Vila do Conde, Portugal; jpd@esmad.ipp.pt
3. CITIC, Research Center of Information and Communication Technologies, Talionis Research Group, Universidade da Coruña, 15071 A Coruña, Spain; javier.pereira@udc.es
* Correspondence: pvg@ess.ipp.pt

**Featured Application: The integration of biofeedback systems in Emotionally Adaptive Immersive Environments contributes to increase their interactivity, allowing them to be used in therapeutic programs.**

**Abstract:** The interactivity of an immersive environment comes up from the relationship that is established between the user and the system. This relationship results in a set of data exchanges between human and technological actors. The real-time biofeedback devices allow to collect in real time the biodata generated by the user during the exhibition. The analysis, processing and conversion of these biodata into multimodal data allows to relate the stimuli with the emotions they trigger. This work describes an adaptive model for biofeedback data flows management used in the design of interactive immersive systems. The use of an affective algorithm allows to identify the types of emotions felt by the user and the respective intensities. The mapping between stimuli and emotions creates a set of biodata that can be used as elements of interaction that will readjust the stimuli generated by the system. The real-time interaction generated by the evolution of the user's emotional state and the stimuli generated by the system allows him to adapt attitudes and behaviors to the situations he faces.

**Keywords:** mental health and wellness; affective computing; empathy; immersive environments; augmented reality; virtual reality; electroencephalography; biofeedback; affective feedback

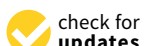



## 1. Introduction

### 1.1. Emotions in the Empathy Construct

Emotions are related to the stimuli that triggered them within a given context [1]. Emotions are part of the empathy process that reflects the ability to share another person's affective state [2,3]. Exposure to immersive environments provides impactful experiences capable of generating different types and intensities of emotions [4]. The emotions generated by an immersive environment designed with a specific purpose can contribute to increase the degree of empathy. Emotions arise through the feeling of realism and presence that involve the user during the exhibition.

The emotional state created by the immersive environment increases receptivity and promotes reflection. A more engaging environment will have a greater influence on the user's perception during the exposure, awakening their senses, generating emotional reactions to the triggered events and creating a sense of realism and presence [4].

The sensation of immersion is caused by a set of factors such as the appearance and graphic quality, the narrative, sound design and interactivity. The interactivity of an immersive system is a process triggered by the emission of different types of stimuli to induce reactions in the user. The stimuli explore the user's senses, the most used stimuli are visual, sound and tactile. Traditionally, these types of stimuli cause voluntary and involuntary reactions (Figure 1).

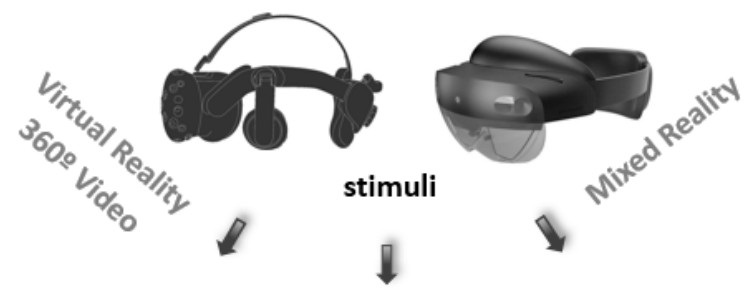

**Figure 1.** Voluntary and involuntary reactions to stimuli in immersive environments.

In voluntary reactions, the user responds more or less consciously to stimuli in order to adapt and influence the environment. The speed and type of response depends on the user's attention and physical and intellectual abilities. Involuntary reactions cause unintended physical reflexes, which can be somehow controlled or inhibited by the user and also induce biological changes over which the individual has no control (unless he has specific training that can influence those changes). Some of the relevant biological changes are reflected by changes in respiratory rate, heart rate and intensity, skin conductivity and through brain waves.

The interactivity of a system is characterized by the type of interaction between the system and the user. The stimuli sent by the system or by the user trigger responses that induce actions that can produce new stimuli.

A truly immersive system explores the potential for voluntary and involuntary reactions. Voluntary reactions can be captured using buttons, movements or eye tracking devices, but to capture certain involuntary reactions, it is necessary to use biofeedback devices. Real-time biofeedback allows to capture and evaluate the user's reactions to stimuli. The use of real-time biofeedback devices should, as far as possible, be non-invasive to avoid interfering with or limiting the immersive experience.

The use of real-time biofeedback mechanisms can add significant value to the interaction process, consequently increasing the feeling of immersion. In this way, an affective algorithm can be used for emotion recognition, identifying and quantifying the emotions felt by the user.

The data collected directly from each biofeedback device, by itself, do not add value to the interaction process if they are not interpreted. These unimodal data must have a first interpretation and conversion, and after this phase, together with the other data, they are

transformed into multimodal data. This process is carried out by an affective algorithm, whose function is to collect, interpret, convert and combine the biodata from the different devices. The affective algorithm is also responsible for analyzing the multimodal data, recognizing emotions, comparing the values with the defined objectives and determining the configuration parameters, defining the types and intensities of new stimuli to be emitted, so that the system can automatically generate interactivity.

In the interactive process in a truly immersive system, the user can respond voluntarily to stimuli when their responses and options are made consciously, but the user can also interact in an unconscious way, having no control over the response to stimuli, which is determined through the biodata collected in real time.

The fact that the self-control of a person's biological state can be developed with time and experience [5] opens doors so that this type of system can be applied in immersive environments for therapeutic purposes, as situations in which it is intended that the user trains self-regulatory mechanisms in the face of concrete situations (phobias, vertigo, anxiety, crisis and catastrophe, etc.).

The objective of this work is to describe an adaptive model for managing data flows generated by bio stimuli for affective algorithms in the design of interactive emotionally adaptive immersive environments, analyzing the interactive biofeedback process as a key element for the development of truly immersive environments.

The Model presented in this work aims to facilitate the planning, design and development of truly immersive interactive environments. Its conceptual base allows its adaptation to different immersive environments (Virtual Reality, Mixed Reality, 360° Video), with different application purposes (therapeutic, self-regulation, training).

### 1.2. Interactivity as One of the Essential Factors for Creating Immersive Environments

To consider the importance of interactivity mechanisms on virtual reality systems and their role on the level of sense of presence and emotions, it is necessary to understand first what means interactivity. This term originates from the term interaction, the relations between human beings [6]. Michael Jäckel explains the sociological concept of interaction, "The basic model on which the sociological concept of interaction is based is the relationship between two or more people who orientate themselves in their behavior and can perceive each other" [7].

Computer sciences used the term interaction to describe the use of computer systems by users. Starting with very basic interfaces to provide that communication between computers and humans, there was an increase in complexity with the advance of the technology. Human–Computer Interaction (HCI) examines the structure of user interfaces and ways of communication using hardware and software. Reference [8] presents three criteria to distinguish the terms interaction and interactivity. "Interactivity" demands for real and observable interactions among humans via a machine or between man and machine, implying real human behavior. Second, interactivity depends on a technical component that occupies a key position within the communication process. Third, no change of devices will be necessary for interactive communication.

In general sense interactivity can be considered to describe an active relationship between two persons or objects [9]. The term gained relevance in the late 1980s and 1990s with the increasing importance of multimedia and several authors reflected and tried to construct a definition. Reference [10] defines interactivity as the "extent to which users can participate in modifying the form and content of a mediated environment in real time". Reference [11] addresses the question of whether an interactive artwork can be immersive (narrative as virtual reality).

Sherman and Craig [12] defined the four dimensions they consider relevant for a VR experience: consumer inhabitation, feelings of immersion, sensory feedback and interactivity. According to [13,14] interaction mechanisms have a significant role on increasing the level of telepresence providing social, cognitive and physical engagement.

### 1.3. Neuro and Biofeedback Systems

Emotions are present in humans' daily life; they are crucial to communicate and to everyday interpersonal events [15,16]. Emotions are a human sentimental state that are a response to an external or internal stimulus, such as a situation, an object or a memory of a past emotional event [16–19]. Humans use emotions to communicate between them and it is important to point out that the emotional state of the person can affect several daily factors, such as learning, making decisions and memory [15,17].

Emotion is an episode of interrelated and synchronized change in five subsystems (Figure 2): cognitive processing, subjective feeling, action tendencies, physiological changes and motor expression [16,18].

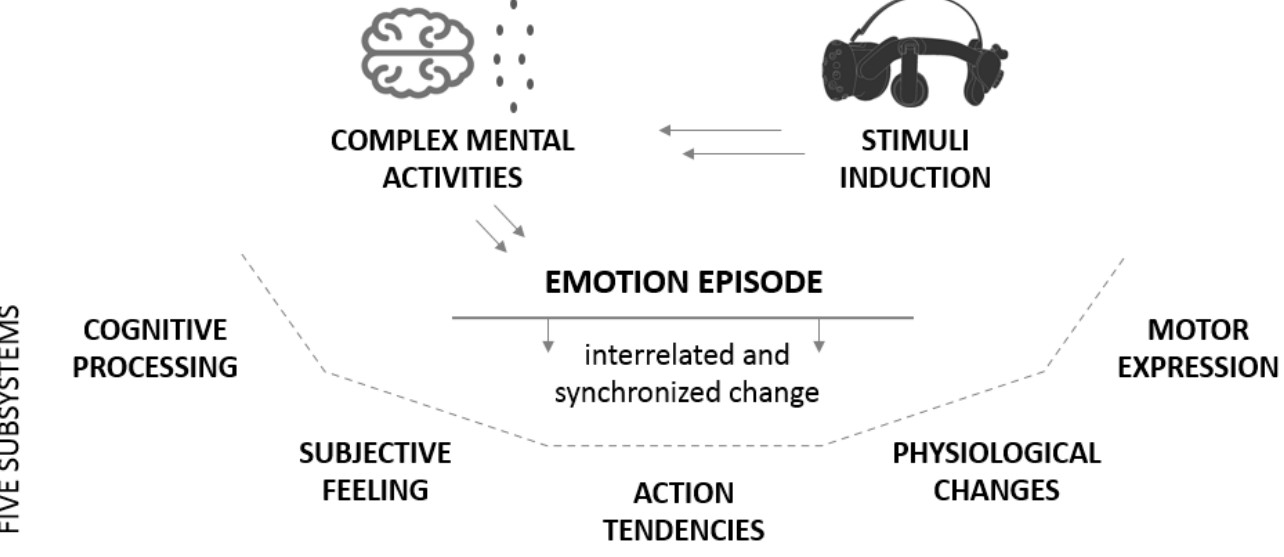

**Figure 2.** Subsystems of interrelated and synchronized change in an Emotion episode.

Emotions comprise complex mental activities, although there is no consensus on a precise and complete definition of emotions [16,18]. Although there is a lot of difficulty in defining emotions, they can be described through two perspectives, the dimensional model and the discrete model (Figure 3). In the dimensional model, there is a need to have two dimensions to describe an emotion, valence (that ranges from unpleasant to pleasant) and arousal (that goes from not aroused to excited). Regarding the discrete model of emotions, the emotions can be discrete as basic emotions or complex emotions, being the last one a combination of basics. The most frequent basic emotions are happiness, sadness, anger, disgust, fear and surprise [16,20,21].

Therefore, nowadays, the ability to recognize emotions is a very important skill to achieve intelligent and effective social communications [22]. The recognition of the different emotional states is very important to several areas of activity, such as medicine, education, intelligent system and human-computer interaction, among others [15,22]. Self-reporting of emotions is unreliable; there are other ways to recognize emotions. It is possible to identify emotions through facial expressions and voice intonation and is also possible to detect emotions via physiological signs, such as electroencephalogram (EEG), cardiovascular changes (heart rate, blood volume pressure and peripheral vascular resistance), respiration patterns, galvanic skin response, skin temperature and body language [15,16,21,22].

The process of emotions involves different parts of the central nervous system which interact between them, especially between the central and autonomic system. There are brain pathways between various cortex areas that have connections to process the emotions [23]. For example, the hypothalamus controls part of the autonomic nervous system responsible for some changes such as heart rate, skin temperature and respiration

rate. The hypothalamus also controls parts of the amygdala, where it is processed the feedbacks from the external environment and also mediates emotional responses [21,22]. EEG can be used to measure and monitor changes in the brain, and it has the advantage that the physiological signals are difficult to tamper with. For that reason, the use of EEG to recognize emotions has been receiving attention from the researchers of this area of expertise [15]. On the EEG analysis, it is necessary to consider some characteristics such as time domain, frequency domain and time-frequency domain of the EEG to correlate the information between the different EEG channels and to find more reliable information; however, these domains can also give information about emotion in separate [15,17]. The time domain can identify characteristics of time and can help to do EEG statistics such as mean, power and standard deviation. The frequency domain is used in some frequency domain techniques such as power spectral density [15].

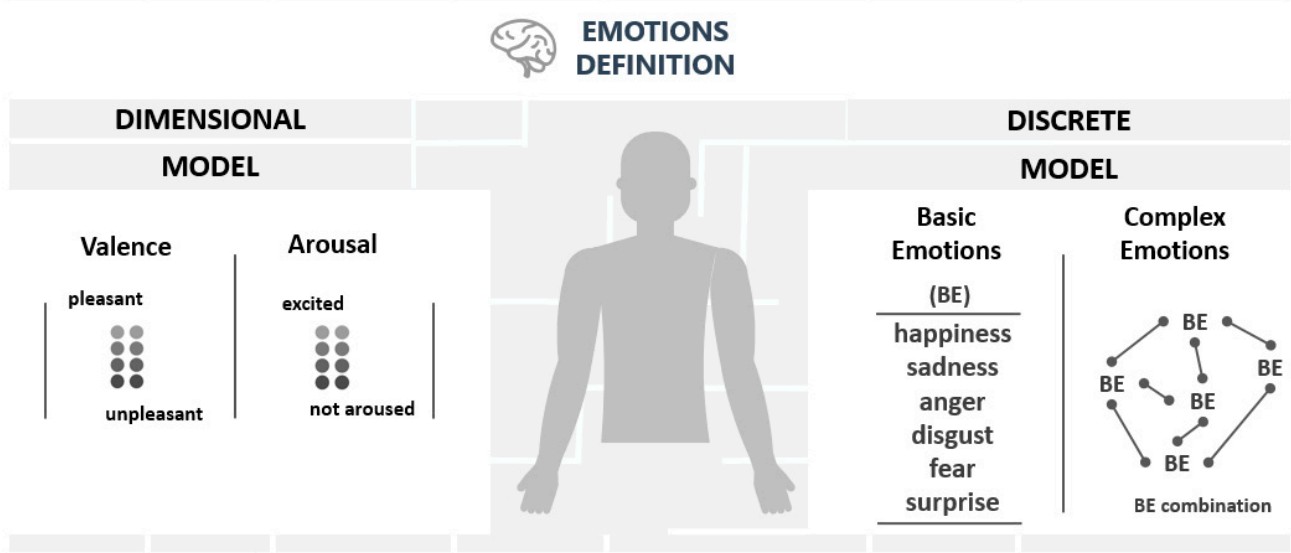

**Figure 3.** The dimensional model and the discrete model in emotions definition.

A good quality EEG is dependent on the disturbances during the process of the recording. It is important to know the characteristics of each artefact to decrease it, or during the recording, removing the disorder, or during the analysis, using signal filters [24,25].

The data after being detected and acquired need to be correctly archived, so in that way, it can be processed and analyzed afterwards. It is important that the storage of these information be done in a general format that can be read by different signal software, so then, it can be processed and analyzed [24].

The EEG signals are also used as a brain–computer interface (BCI), which can communicate between a computer and the individual, enabling the human brain to control external equipment. The BCI technology can be used in different areas, like psychology, medicine and neurogaming [26,27]. In neurogaming, it is possible for the user to control the video game with the brain waves. These kinds of games can have therapeutical aims, for example, for improving the working memory. There are studies that use BCI based on non-invasive EEG with virtual and augmented reality. Using these technologies together gives the participant a greater sense of immersion [26,28].

Although an agreement does not yet exist on the patterns and the brain regions that are responsible for the emotions, regardless of the subject, it is believed that the amygdala is either responsible for fear or that is one of the most important brain areas to process this emotion. This part of the brain is most likely to be active when the person cannot predict what sensations can mean, what to do about it or if they are valuable or not in that specific situation [21,29]. The brain activation is considered when there is a decrease in alpha EEG power [30]. There is a study that found evidence that the posterior cingulate

cortex, precuneus and medial prefrontal cortex are representative of emotions; the stimuli suggest emotional information or memories of an emotional stimuli. The angular gyrus is an important component in the emotional memory retrieval [19]. Lindquist et al. (2012) found that the brain regions that are involved in basic psychological operations, being emotional or not, are active during the emotion experience [29].

Cortex electrical activity is produced tanks to the activation of the neurons of the cortex. This activity is recorded in the scalp where those neurons are located. This is complex and variable which transmit irregular signals to the computer screen. Despite this irregularly and variability, it is possible to recognize patterns that can be divided into several waves that are identified using frequency (Hertz—Hz) and amplitude (microvolt—μV), such as Alpha (8–13 Hz), Beta (13–30 Hz), Theta (4–8 Hz), Delta (<4 Hz) and Gamma (30–100 Hz). These frequencies also represent different physiologic function, for example, delta waves are seen during sleep, while theta waves are seen during the sleepy stage. When a person is relaxed but awake, this is seen alpha waves, meanwhile when that person is alert, the EEG shows beta waves, and during problem solving, gamma waves are seen [25,31,32].

Another EEG feature that is very important in emotion recognition is the frontal asymmetry and midline power. Regarding the midline power, it can be associated with emotional processing, especially the frontal midline theta, which was associated with the relaxation state from anxiety and positive emotional events [16].

The asymmetric brain activity in the frontal area between the right and left hemispheres is responsible for frontal asymmetry. The frontal asymmetry is normally seen in alpha (8–13 Hz) and theta (4–8 Hz) rhythms and is a potential tool to recognize discrete emotions. For example, anger can provoke higher levels of left frontal activity and low right frontal activity, but the frontal asymmetry is also related to valence, arousal and self-reported dominance [16]. Another way to determine the asymmetry is using the spectral asymmetry index [33]. Orgo et al. (2015) found that negative stimuli increased the spectral asymmetry index in frontocentral, central, centroparietal, parietal and occipital areas, compared to the neutral stimuli. They also found evidence of significant decrease of the spectral asymmetry index in left temporal, centroparietal, parietal and occipital areas when the person was subject to positive stimuli [33]. It was also found that the frontal right hemisphere is more active in response to negative emotions, and in the frontal left hemisphere, there was an increase of activity with positive emotions [30,34]. There are two hypotheses to explain these findings. The first theory explains that the right hemisphere is dominant in the process of emotions regardless of the affective valence. The second theory justifies that the right hemisphere is responsible for processing the negative emotions, and the left hemisphere is specialized in the positive ones [23].

The frontal power asymmetry is a measure frequently used in neurofeedback in order to train emotional self-regulation [30]. Neurofeedback is a non-invasive technique that is an application of brain–computer interface, and it is being used in the treatment of mental disorders as well enhancing brain performances such as behavior, cognitive and emotional processes, in real time [30,35]. Neurofeedback is a technique that has been gaining attention because in addition to the benefits that are known to the patient, it also guides mental activity from the brain area in study through real time feedback [30,35,36].

The subject can learn how to improve the brain activity; although besides the usual applications in 2D, there are studies that use neurofeedback on 3D immersive environments [37,38]. 3D environments offer more user's interaction with the world around them which helps in the simulation of real-world tasks [38]. The characteristics inherent to the immersive environment make the users feel that they are really there and give them a sense of presence, since it is possible to create complex and real situations and/or environments [39,40].

Due to the sense of presence that this technology gives to the user, it has been indicated as a tool for provoking emotions in laboratory environments. There are several studies that show that immersive environments can provoke different emotional states like anxiety,



relation, and different mood in social environments featuring avatars [38,39]. Immersive environments are also a tool for behavioral research in psychological assessment [39].

Electroencephalography is a painless and non-invasive technique to acquire electrical activity from the brain, using electrodes on standard positions on the head that conduct the electrocortical activity to the amplification equipment, which is necessary since this cerebral cortex information is measured in microvolts and needs to be amplified to be exhibited on a computer [24,25,41]. There are several advantages of EEG for the diagnosis of different pathologies, such as, the sensitivity of EEG, almost real-time recording, the accessibility to the recording and EEG moderately low cost [41,42].

Besides other applications of 3D environments like education or architecture, it has been proven as an effective technology for therapeutic applications, since 3D environments have stronger effects than 2D stimulation [39,43].

## 2. Biofeedback for Self-Regulation Training in Adaptive Environments Model

Exposure to immersive environments affects the user by creating sensations that arouse emotions, causing involuntary reactions such as changes in heart rate and intensity in respiratory rate, brain activity, skin conductivity and eye movements [44]. There are biofeedback devices capable of recording these changes that occur during exposure. This biofeedback allows the analysis of the impact that the immersive environment had on the user, but it is possible to go even further; real-time biofeedback can be used as an element of interactivity.

The biodata captured by biofeedback devices allow to determine how the stimuli affect the user; the conversion of unimodal data obtained by each device into multimodal data allow, through an affective algorithm, to understand how mental changes influence the user's emotional state.

The multimodal data are interpreted by the affective algorithm. In a first phase, it uses the Discrete Model to identify the emotions felt by the user, and in a second phase, it uses the Dimensional Model to determine the valence and arousal of the emotions under analysis. The combination of these two models allows to determine the emotional state of the user through the binomial emotion-intensity.

The affective algorithm identifies the user's emotional state in real time and can use two interactivity strategies, a passive interactivity strategy (PIS) or an active interactivity strategy (AIS). The difference between these two strategic approaches is in the information that is provided in real time to the user (Figure 4).

The passive interactivity strategy, after identifying emotions and quantifying their intensity, uses these values for the system to generate new stimuli that can lead the user to the emotional state that is intended. In PIS, the user does not have access to information about his own emotional state, and the interactivity is entirely controlled by the system.

The active interactivity strategy provides the user with real-time information about his own emotional state, allowing him to regulate his emotions to control the stimuli sent by the immersive environment. In AIS, the user is aware that his behavior influences the system, and through self-regulation processes, he can train which responses are most appropriate to the stimuli generated by the system.

Both passive and active interactivity strategies, through measurement of user emotional data, allow to create an Emotionally Adaptive Immersive Environment which continuously adapts the stimulus to the user emotional state. The continuous adaptation of stimuli to the user's emotional state makes each exposure a unique and personalized experience. During the exhibition, adaptive behavior influences the user's attitudes and his response to stimuli (Figure 5).

When the participant receives information in real time about a certain aspect of his physiology during an exposure, or a sequence of exposures, to an immersive environment, the participant may realize that changes in his mental state can influence the environment. This progressive awareness influences and transforms the participant's attitude. (Figure 5).

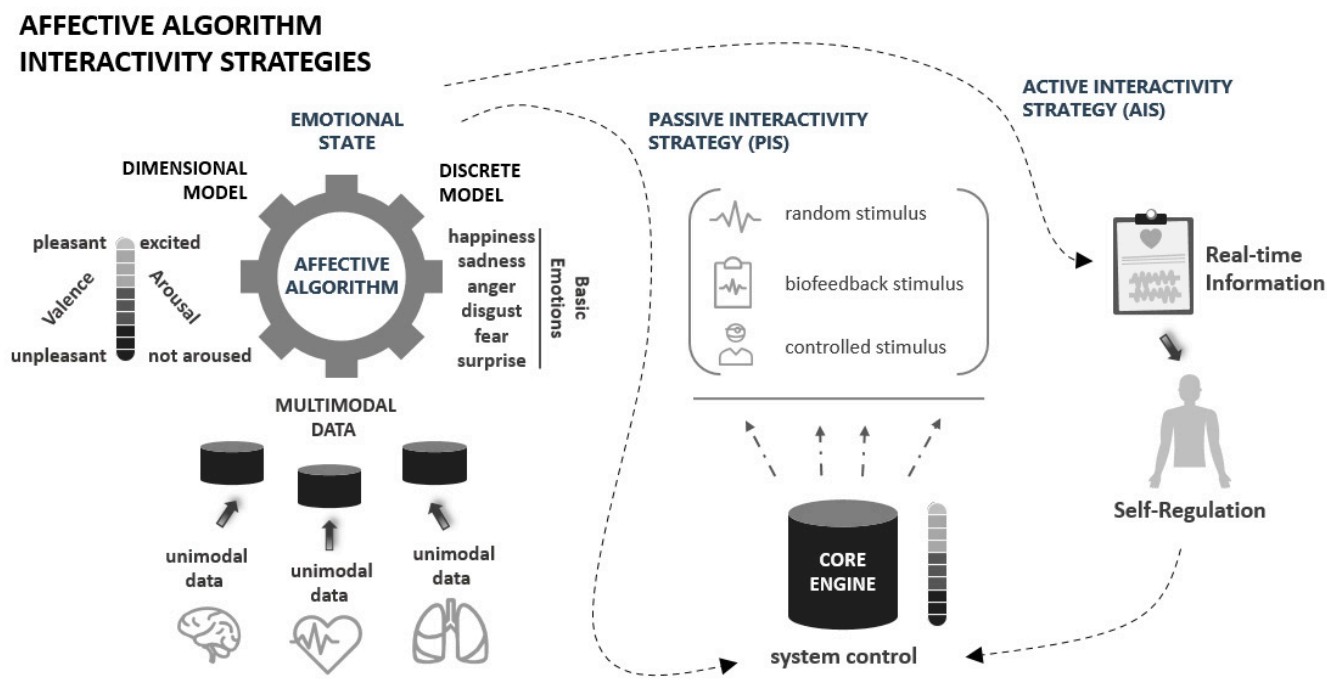

**Figure 4.** Passive and Active Interactivity strategies used by Affective Algorithms in immersive environments.

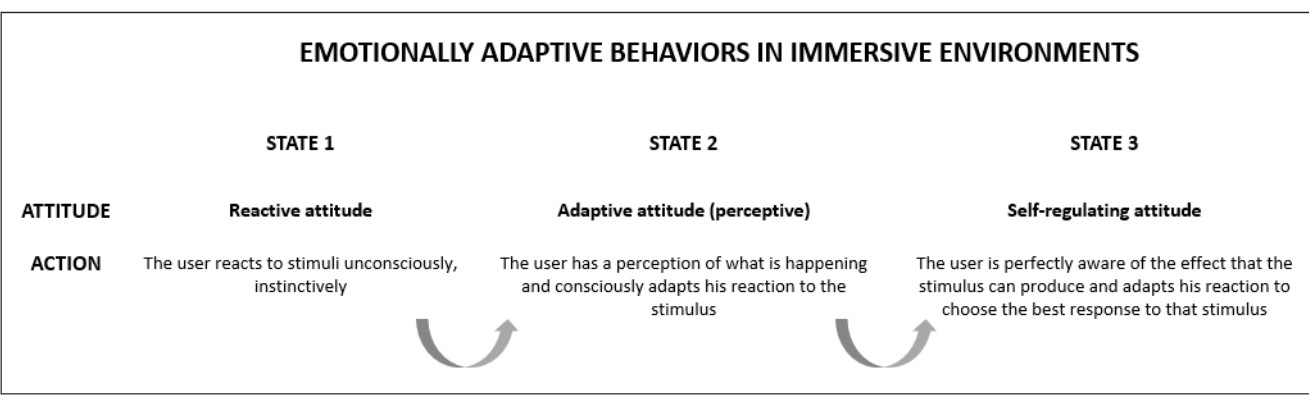

**Figure 5.** Behavioral change during continuous exposure to stimuli generated in immersive environments.

A biofeedback system applied in an immersive environment transforms it into an emotionally adaptable and immersive environment, where the user's emotional state can optimize the experience through the continuous adaptation of stimuli to his own emotional state.

The Emotionally Adaptive Immersive Environment uses the affective algorithm to relate stimuli to emotions (Figure 6). The system can generate different stimuli, visual, audible or tactile; their impact depends on their frequency, intensity and duration.

The stimuli trigger voluntary and involuntary reactions in the user. Voluntary reactions reflect a conscious response; the user chooses the attitude considered most appropriate to respond to the stimulus, having control over his emotional state.

Involuntary reactions are manifested through physical reflexes and biological changes. The movement of the eyes and deviating or shrinking the body are examples of physical reflexes that naturally happen in unexpected situations. These movements can be captured through sensors that inform the system of the position that the user takes, and they are also important elements in the interactivity process.

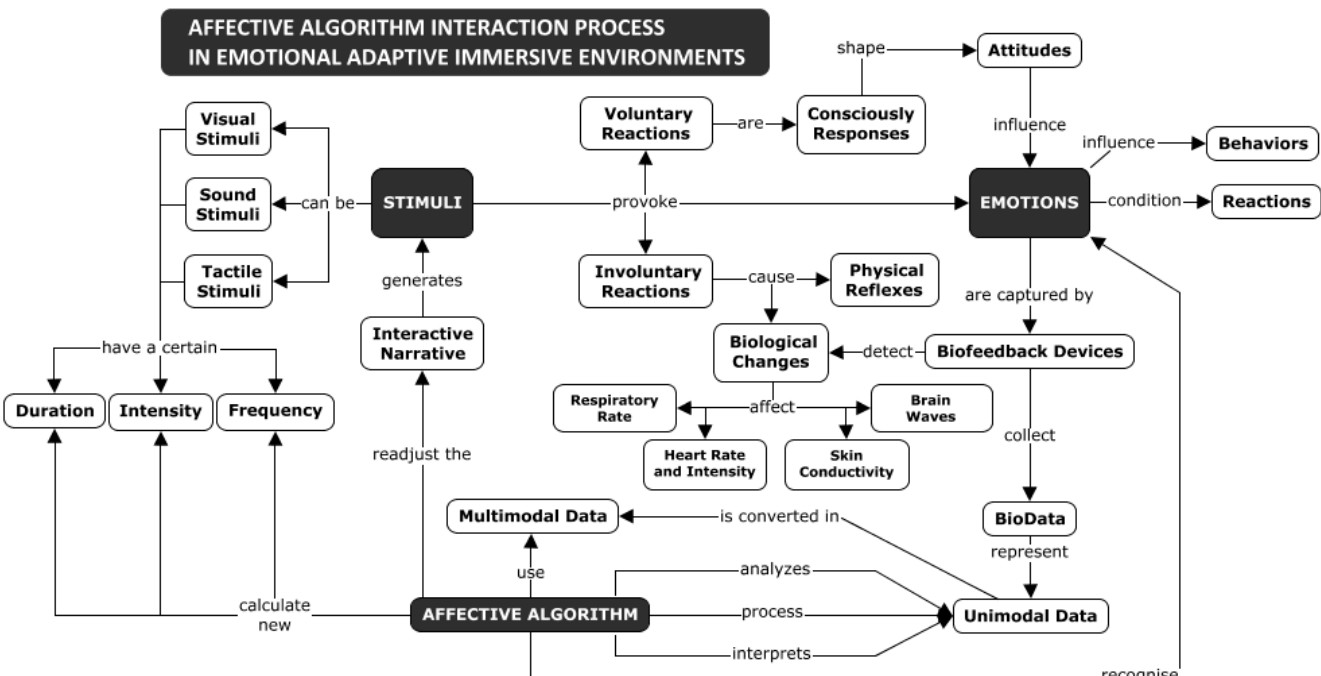

**Figure 6.** Biofeedback mechanisms to identify emotions and generate interactivity in adaptive immersive environments conceptual protocol model.

Biological changes as respiratory rate, heart rate and intensity, skin conductivity and brain waves can be captured, in real time, using biofeedback devices. Those biological changes are involuntary, and their intensity may vary from person to person. Each of them generates data, the unimodal data. Unimodal data are extremely relevant, as they characterize the biological response that the human body gives to each stimulus received. The analysis, processing and interpretation of the collected unimodal data allows its conversion into multimodal data (Figure 7).

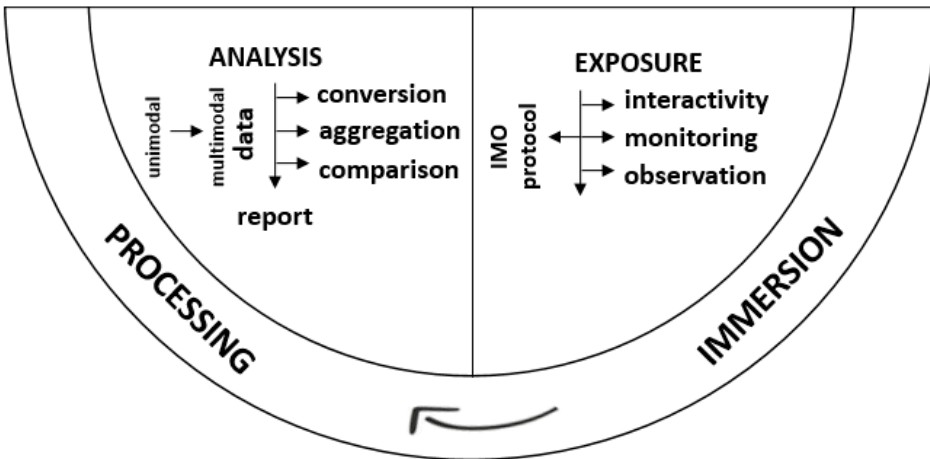

**Figure 7.** Affective algorithm in the conversion from unimodal data to multimodal data after application of the IMO protocol.

The affective algorithm uses multimodal data to recognize emotions. This process makes it possible to determine the user's emotional state during exposure to stimuli and is a key element in the affective algorithm interaction process in emotionally adaptive immersive environments. The use of multimodal data makes it possible to assess the type and intensity of emotions felt with a greater degree of reliability.

The affective algorithm, in addition to analyzing and interpreting the multimodal to determine the user's mental state, compares these values with standard values and calculates new values for the frequency, intensity and duration of the stimuli, which allow to reach the intended objectives. In this way, the affective algorithm has a real-time influence on the exhibition, transforming the narrative itself into an interactive narrative.

## 3. Materials and Methods

### 3.1. Materials

Non-evasive electroencephalogram devices (Looxid Link™ Mask for VIVE) can be used coupled with VR Headsets like HTC Vive Pro™ or the VR Headset HTC Vive Pro Eye™ with Precision eye tracking capabilities (Table 1). Looxid Link™ Mask for VIVE device is compatible with those VR Headsets, and its use does not interfere with the immersive experience because the participant does not realize that he is using it.

**Table 1.** EEG equipment specification applied to Emotionally Adaptive Immersive Environment.

| Equipment | Specifications |
|---|---|
| Computer | CPU: Intel® Core™ i7-9700K (3.60 GHz–4.90 GHz)<br>Graphic card: NVIDIA® GeForce® RTX 2080 Ti<br>Memory: 64 GB RAM |
| VR Headset<br>HTC Vive Pro™<br>Or<br>Vive Pro Eye™ | High resolution Dual AMOLED 3.5″ diagonal screens<br>$1440 \times 1600$ pixels per eye ($2880 \times 1600$ pixels combined)<br>Refresh rate: 90 Hz<br>Field of view: 110 degrees<br>Integrated microphones with 3D Spatial Audio<br>Four SteamVR Base Station 2.0: $10 \text{ m} \times 10 \text{ m}$<br>VIVE Wireless Adapter |
| Looxid Link™ Mask for VIVE | EEG sensors<br>Looxid Link Hub<br>6 channels: AF3, AF4, AF7, AF8, Fp1, Fp2<br>1 reference: FPz at extended 10-10 system<br>Dry electrodes on flexible PCB<br>Sampling rate: 500 Hz<br>Resolution: 24 bits per channel (with 1 LSB = 0.27 μV)<br>Filtering: digital notch filters at 50 Hz and 60 Hz, 1–50 Hz<br>digital bandpass<br>Real-time data access<br>Raw EEG data: 500 Hz (with/without filter options)<br>Feature indexes (alpha, beta, gamma, theta, delta): 10 Hz<br>Mind indexes (attention, relaxation, balance): 10 Hz |

Other data acquisition and analysis systems such Biopac™ MP160 (Table 2) allow to capture biological signals (cardiovascular changes, respiration patterns, galvanic skin response) in a non-invasive way and without disturbing the sensation of immersion, maintaining the freedom of movement of the participant, since they can transmit data wirelessly, which avoids the discomfort and inconvenience of cable connection.

**Table 2.** ECG, plethysmography and galvanic skin response equipment specification applied to Emotionally Adaptive Immersive Environment.

| Equipment | BIOPAC™ MP160—Specifications |
|---|---|
| ECG | Number of Channels: 16<br>Absolute Maximum Input: ±15 V<br>Operational Input Voltage: ±10 V<br>A/D Resolution: 16 Bits<br>Accuracy (% of FSR): ±0.003<br>Input impedance: 1.0 MΩ<br>Amplifier Module Isolation: Provided by the MP unit, isolated clean power<br>CE Marking: EC Low Voltage and EMC Directives<br>Leakage current: <8 μA (Normal), <400 μA (Single Fault)<br>Fuse: 2 A (fast blow) |
| ECG and Respiratory Amplifier | Transmitter: Ultra-low power 2.4 GHz bi-directional digital RF transmitter<br>Rate: 2 kHz, maximum<br>Screen: Color, 6 cm diagonal RF reception range: 1 m (line of sight, approx.)<br>Memory: 32 GB<br>Built-in Accelerometer: X, Y, Z—axes; rate 100–400 Hz; Range: ±2–16 G |
| Plethysmography and galvanic skin response | Signal type: PPG plus EDA<br>Resolution: PPG: FSR/4096, (4.88 mV); EDA: 0.012 μS (min step)<br>Operational range: 10 m<br>Transmitter: Ultra-low power, 2.4 GHz bi-directional digital RF transmitter;<br>Rate: 2.000 Hz (between transmitter and receiver) |

### 3.2. Methods

Interactivity, as one of the factors that trigger the sensation of immersion, must be considered in the development process of immersive environments. The interaction generated by involuntary reactions must be carefully studied, as well as the interaction that uses voluntary reactions.

The authors propose an adaptive model for biofeedback data flows management, to manage the data generated by the participant's involuntary reactions, to be incorporated in the requirements specification in the design of interactive immersive systems.

#### 3.2.1. Biodata as Elements of Interactivity

To select which involuntary responses to stimuli are most appropriate for use as an interaction element, several aspects must be considered:

- The purpose of exposure to the immersive environment.
- The role of the participant.
- Identifying the possible types of reactions.
- The mobility allowed to the participant during the exhibition.

The selection of the biodata used as an interaction element must be aligned with the purpose of the exposure. Brain activity, cardiovascular changes, galvanic skin response, skin temperature, respiratory rate and eye movements result from exposure to stimuli and are generators of biodata that can be used as an interaction element. The selection of the types of biodata to be used is decisive for the choice of the most appropriate equipment.

A truly immersive system uses an adaptive environment and explores the participant's involuntary reactions as an element of interaction. The participant can have a more active or more observant role.

The identification of possible reactions allows to define the division of the scales of values of the biodata in intervals of values. The parameterization of the affective algorithm defines the number of intervals on each scale and the range of values for each interval, to assign the degree of sensitivity of the system to the reactions of the participant.

One of the factors to consider during the exhibition is the mobility of the participant: the participant's position (standing, sitting), the need to move in space, the possibility of

performing sudden movements and the proximity to electrical or magnetic equipment that may interfere with the devices for collecting the biodata.

### 3.2.2. Interactive Immersive Environments

To test the model presented, the authors developed two immersive environments as similar as possible and able to use the same core engine. The first immersive environment in Virtual Reality was designed with the aim of reproducing, using Virtual computer graphics scenarios, the same room that was used in the second immersive environment that was developed using 360° video.

The same 360° narrative was applied to both environments. Each exhibition lasts approximately seven minutes, with the aim of simulating the realization of a Stroop test that integrates the process of selecting recruitment for a job.

During the exhibition, the immersive environment generates a series of visual and sound stimuli, which simulate the hallucinations that a person with schizophrenia experiences during a psychotic disorder.

The participant must be focused to perform the Stroop test despite the stimuli to which he is being exposed. On the other hand, the participant's emotional state influences the amount and intensity of stimuli generated by the system. The anxiety and excitement revealed by the participant generate more distracting stimuli, and the concentration and focus on the task decrease the emission and intensity of the stimuli generated by the system, inducing self-regulating behavior.

This exposure makes the participant experience, for a few minutes, the same sensation that a person with schizophrenia feels during a psychotic disorder, understanding the effect that this type of flare can have on the performance of daily tasks.

Biodata selected for this experience used as an interaction element:

- Brain activity.
- Cardiovascular changes.
- Galvanic skin response.

Participant role during the experience:

- Active role.

Possible reactions:

- During the experiment, an expert monitors the reactions of the participant.
- The experience can be interrupted at any time by the participant's initiative.
- The expert may interrupt the experiment if he considers that the participant's reactions endanger safety.

Position and mobility of the participant:

- Sitting on a chair.
- Despite being in a fixed position, the participant can freely move his arms, torso and head in order to explore the surrounding environment.

The system control Core Engine is based on real-time Unity platform. The Core Engine (Figure 6) receives the biodata from the devices connected to the participant and analyzes, processes and interprets the values, converting the unimodal data into multimodal data through the IMO protocol (Figure 7). During the exposure, the system readjusts the stimuli in real-time according to the values returned by the affective algorithm. It was applied to the two immersive environments developed with exactly the same narrative, the same timeline, the same stimuli and the same intensities.

## 4. Results

Empathy VR-Schizophrenia is a Project, within the scope of promoting mental health and wellness literacy, which aims to study the impact of using immersive environments for the participant to create empathy towards people with schizophrenia.

The Adaptive Model for Biofeedback Data Flows Management was applied in the design of two Interactive Immersive Environments. To test the model's implementation for real cases, the authors developed two different types of immersive environments (Figure 8), a virtual reality environment (virtual computer graphics scenarios) and an immersive environment using 360° video.

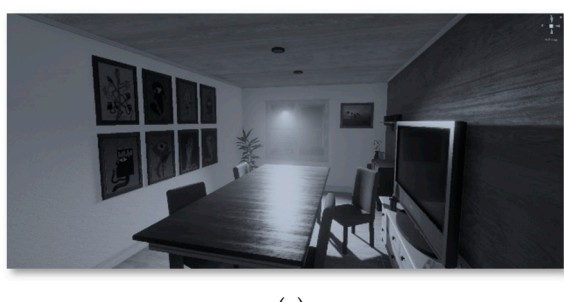 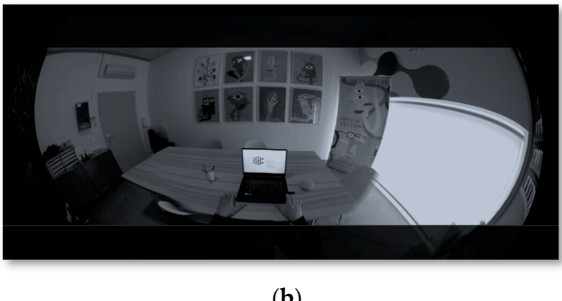

(**a**)                                                                                 (**b**)

**Figure 8.** Empathy VR—Schizophrenia Project Scenarios: (**a**) virtual reality—virtual computer graphics scenario; (**b**) 360° video—real scenario.

To verify the applicability of this model, the authors chose to develop two different immersive environments with similar characteristics, sharing the same purpose and the same narrative. The application of the same model in the development of the two environments made their performance and their dynamic and adaptive behavior similar. The two environments generate the same stimuli and intensities for the same reactions of the participants.

## 5. Discussion

The application of the Adaptive Model for Biofeedback Data Flows Management allowed to conceptualize, design, develop and test two different types of immersive environments.

The conceptualization of an immersive environment requires a deep reflection on several factors that can determine its success. The engineering process to develop this type of system must consider as requirements, in addition to the socio-technical aspects, a strong human–computer interaction component.

One of the important factors is the use of 360° narratives, in which it is not possible to control where and when the participant will direct his gaze. The autonomy granted to the participant to freely explore the surrounding environment means that each participant can have a different experience. As it is not possible to "force" the participant to look at a certain point, the emission of stimuli is the way to get their attention.

The use of equipment to collect biodata implies mobility restrictions, and these restrictions must consider the movement through space and the performance of sudden movements. The ranges of values collected by the biofeedback devices must consider their sensitivity.

The developed environments had the purpose of creating empathy towards people with schizophrenia; the application of this adaptive model, allowed to develop two immersive environments, using different technologies, but ensuring that both met the defined objectives and requirements.

## 6. Conclusions

The reaction to stimuli can vary from person to person, and the same person may react differently to the same stimulus in different situations. Exposure to immersive environments contributes to increasing the receptivity for the construction of emotions. There is a relationship between emotions and the stimuli that triggered them. The emotional state influences the type of reaction and the ability to respond to stimuli.

The emotions that the participant feels during exposure to immersive environments trigger voluntary and involuntary responses, conditioning their actions and attitudes. Faced with a stimulus, the participant may have adaptive, reactive, or self-regulating attitudes.

During exposure to immersive environments, biofeedback devices can be used for monitoring and observation. The collection and processing of biodata generated by biofeedback devices in real time allows the use of biodata as an element that generates interactivity.

An Emotionally Adaptive Immersive Environment can use an affective algorithm to convert unimodal data into multimodal data to recognize emotions and relate them to the stimuli that caused them. The mapping between stimuli and emotions made by the affective algorithm allows to raise the level of interactivity of the system to another level.

The type of interactivity that real-time biofeedback systems provide allows their use in therapeutic programs, among which stand out those that promote adaptation behaviors or self-regulation in the face of uncomfortable situations for the user. This investigation opens the possibility of using this type of Emotionally Adaptive Immersive Environments in different areas, which can be integrated into therapeutic programs related to the treatment of some types of phobias (for example, arachnophobia or social phobia) and programs for self-regulation of behaviors (anxiety control, improve the ability to concentrate).

**Author Contributions:** Conceptualization, P.V.G.; investigation, P.V.G., A.M., J.D., C.S. and A.C.; methodology, P.V.G. and A.M.; project administration, P.V.G.; supervision, A.M. and J.P.; validation, P.V.G., A.M. and J.P.; visualization, P.V.G. and A.C.; writing—original draft, P.V.G., J.D. and C.S.; writing—review and editing, P.V.G., A.M. and J.P. All authors have read and agreed to the published version of the manuscript.

**Funding:** This publication cost was funded by CITIC, Research Center of Information and Communication Technologies, University of A Coruña.

**Institutional Review Board Statement:** Not applicable.

**Informed Consent Statement:** Not applicable.

**Acknowledgments:** This research was carried out and used the equipment of the LabRP-CIR, Psychosocial Rehabilitation Laboratory, Center for Rehabilitation Research, School of Health, Polytechnic Institute of Porto. This work had scientific support from CITIC, Research Center of Information and Communication Technologies, University of A Coruña.

**Conflicts of Interest:** The authors declare no conflict of interest.

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
