# Peer review of "Adaptive Model for Biofeedback Data Flows Management in the Design of Interactive Immersive Environments"

_applsci, doi:10.3390/app11115067_

Round 1

Reviewer 1 Report

Undoubtedly, the interactivity of an immersive environment comes up from the relationship that is established between the user and the system. This relationship results in a set of data exchanges between human and technological actors. In this article Authors describes an adaptive model for biofeedback data flows management used in the design of interactive immersive systems. 

My comments to the article are as follows:

- As part of Introcution, I propose to extend the Introduction section by referring to the existing implementations in the field of EEG correlation with AR and VR. For example, I propose a reference to: Augmented Reality of Technological Environment in Correlation with Brain Computer Interfaces for Control Processes, Recent Advances In Automation, Robotics and Measuring Techniques, Book Series: Advances in Intelligent Systems and Computing, Springer from 2014 and Using BCI and VR Technology in Neurogaming, Analysis and classification of eeg signals for brain-computer interfaces, Book Series: Studies in Computational Intelligence, Springer from 2020.

In addition, I propose to expand the information in the Introduction to the article in the field of a brief description of the methods of acquisition, archiving and analysis of the EEG signal. Here, for example, reference can be made to: Methods of Acquisition, Archiving and Biomedical Data Analysis of Brain Functioning, Biomedical Engineering And Neuroscience, Book Series: Advances in Intelligent Systems and Computing, Springer from 2018.

- There is no clearly separated Materials and Methods section within the article. This should be completed.

- In addition, you must enter a Results section in the article. Then you have to add the Disscusions section. Then it will be possible to fully evaluate its scientific aspect.

- As part of the Conclusions, plans for the future regarding the conducted research should be completed.

Author Response

Dear Reviewer,

thank you for your valuable contribution.

We send a new version of the article, in word format, with the change log activated, the added or altered parts are in a different color for easy reference and reading.

All suggestions (content, bibliography and structure) were considered. we hope to have met expectations.

Reviewer 2 Report

The paper subject is relevant and the proposed submission has interesting for readers of the scientific community. The abstract is reflected the paper content . The introduction is good, but it should be extended. In section 1, the authors enumerate many research reviews, which are related to this paper? what is the motivation and significance of this paper? Please extract the motivation from above researches. One of important initial information for the proposed design of Interactive Immersive Environments is EEG signal and other bio-signals. At the present time the analysis of EEG signal is developed intensively. I’d like recommend to consider the study in EEG analysis:
- M. Zhao, H. Gao, W. Wang and J. Qu, "Research on Human-Computer Interaction Intention Recognition Based on EEG and Eye Movement," IEEE Access, vol. 8, pp. 145824-145832, 2020.
- Rabcan, J., Levashenko, V., Zaitseva, E., Kvassay, M. “Review of methods for EEG signal classification and development of new fuzzy classification-based approach” IEEE Access, vol. 8, pp. 189720–189734, 2020.
- H. Peng et al., "Multivariate Pattern Analysis of EEG-Based Functional Connectivity: A Study on the Identification of Depression," IEEE Access, vol. 7, pp. 92630-92641, 2019.
- H. Jebelli, M. Mahdi Khalili and S. Lee, "A Continuously Updated, Computationally Efficient Stress Recognition Framework Using Electroencephalogram (EEG) by Applying Online Multitask Learning Algorithms (OMTL)," IEEE Journal of Biomedical and Health Informatics, vol. 23, no. 5, pp. 1928-1939, 2019. 
The contribution of this work should be presented in more detail:
-    Could you provide some experimental investigations?
-    Could you propose the comparative analysis with alternative design approaches?
-    Could you indicate in Fig.6 the part with is contribution of the authors?
-    The mechanisms in Fig. 6 for the identify emotions and generate interactivity is in general. But some methods and/or algorithms of this mechanisms described in details are welcome. 

Author Response

(The authors gave the same response as above.)

Round 2

Reviewer 1 Report

Dear Authors,

Thank you for the corrected article and for the submitted replies.

All changes have been made.

During the formatting in the version attached by you, point 4 merged with a paragraph of text. Please correct this before final publication.

I recommend the article for publication.

Reviewer 2 Report

I recommend the paper publication. Authors take into account my comments